# Dietary Management of Eosinophilic Esophagitis in the Era of Molecular Diagnostics: The Role and Limitations of Component-Resolved Diagnostics—A Narrative Review

**DOI:** 10.3390/nu17223588

**Published:** 2025-11-17

**Authors:** Adam Wawrzeńczyk, Katarzyna Napiórkowska-Baran, Maciej Szota, Paweł Treichel, Justyna Durślewicz, Zbigniew Bartuzi

**Affiliations:** 1Department of Allergology, Clinical Immunology and Internal Diseases, Collegium Medicum Bydgoszcz, Nicolaus Copernicus University Toruń, 85-067 Bydgoszcz, Poland; knapiorkowska@cm.umk.pl (K.N.-B.); maciejszota98@gmail.com (M.S.);; 2Student Research Club of Clinical Immunology, Department of Allergology, Clinical Immunology and Internal Diseases, Collegium Medicum Bydgoszcz, Nicolaus Copernicus University Toruń, 85-067 Bydgoszcz, Poland; treichel.pawel@gmail.com; 3Faculty of Medicine, Bydgoszcz University of Science and Technology, Aleje Prof. S. Kaliskiego 7, 85-796 Bydgoszcz, Poland; 4Department of Tumor Pathology and Pathomorphology, Oncology Centre—Prof. Franciszek Łukaszczyk Memorial Hospital, 85-796 Bydgoszcz, Poland

**Keywords:** eosinophilic esophagitis, component-resolved diagnostics, elimination diet, precision nutrition, multi-omics, food allergy, biomarkers, patient-centered care

## Abstract

Eosinophilic esophagitis (EoE) is a chronic, immune-mediated disorder characterized by eosinophilic infiltration of the esophageal epithelium, primarily driven by exposure to food and aeroallergens. Although dietary elimination remains the cornerstone of therapy, the identification of specific food triggers still largely relies on empiric methods. This narrative review explores the diagnostic and therapeutic role of component-resolved diagnostics (CRD) and other molecular tools in the personalized management of EoE. Across observational and cohort studies, CRD has shown improved sensitivity in detecting clinically relevant allergen sensitizations compared with conventional tests, allowing for more precise dietary guidance and, in some cases, reducing unnecessary food exclusions. However, remission rates achieved through CRD-guided diets remain comparable or slightly lower than those obtained with empiric elimination, highlighting the need for validation in prospective, controlled studies. Recent advances in omics-based diagnostics, including gene expression profiling and proteomic biomarkers, further underscore the potential of integrating molecular and immunologic endotyping into clinical practice. Overall, current evidence supports CRD as a promising adjunctive tool that enhances the precision of allergen identification but is not yet ready to replace empiric dietary strategies. Future research should focus on validating standardized CRD-guided algorithms, integrating omics-derived biomarkers, and developing non-invasive diagnostic platforms. Incorporating dietitian-led nutritional assessment and biomarker monitoring into CRD- and omics-informed care pathways may help prevent nutrient deficiencies, improve adherence, and translate molecular precision into safer, patient-centered dietary management.

## 1. Introduction

Eosinophilic esophagitis (EoE) is a chronic, immune-mediated disease of the esophagus characterized by symptoms of esophageal dysfunction and dense eosinophilic infiltration of the epithelium. It has emerged as one of the leading causes of dysphagia and food impaction in both children and adults, with a steadily rising incidence across Western populations. The pathogenesis of EoE involves a complex interplay between genetic predisposition, epithelial barrier dysfunction, and type 2 (Th2)-driven immune responses to food antigens. Dietary proteins are central to disease initiation and persistence, and the elimination of specific dietary triggers remains a cornerstone of therapy [1,2,3].

At the molecular level, EoE reflects a disrupted epithelial barrier that facilitates allergen penetration and activation of type 2 immune pathways. Exposure to dietary and environmental allergens induces epithelial release of thymic stromal lymphopoietin (TSLP), interleukin-25 (IL-25), and interleukin-33 (IL-33), which promote Th2 polarization and activate group 2 innate lymphoid cells (ILC2). These cells produce interleukin-5 (IL-5) and interleukin-13 (IL-13), leading to eotaxin-3 (CCL26) expression, eosinophil recruitment, and tissue remodeling with basal cell hyperplasia and fibrosis [4,5,6]. This epithelial–immune crosstalk underpins both the chronicity and relapsing nature of the disease [7].

Dietary treatment strategies in EoE range from elemental nutrition, which ensures near-complete antigen avoidance, to empiric multi-food elimination and, more recently, diets guided by allergy testing or molecular diagnostics. While elemental diets demonstrate the highest efficacy, their practical limitations—particularly poor palatability and adherence—have restricted their long-term use [1,4,5]. Consequently, empiric elimination diets have become the most feasible and widely adopted approach in clinical practice, offering a balance between therapeutic effectiveness and patient acceptability [1,3,4].

Given the nutritional implications of prolonged food restriction, structured dietary supervision and nutrient monitoring are integral to safe and sustainable EoE management. Dietitian-led assessment is particularly important in pediatric and adolescent populations, where growth, bone health, and micronutrient status may be affected [1,2].

In contrast, diets directed by conventional allergy testing, such as skin prick tests (SPT), atopy patch testing (APT), and serum-specific IgE (sIgE), have shown inconsistent results and limited predictive value for identifying causative food triggers [3,4]. This inconsistency has stimulated interest in advanced immunologic and molecular techniques that could refine the personalization of dietary therapy.

To address these limitations, novel molecular approaches such as component-resolved diagnostics (CRD) have gained increasing attention. CRD and multiplex allergen platforms represent a significant step toward precision nutrition by differentiating genuine sensitization from clinically irrelevant cross-reactivity. Preliminary studies, such as those by Armentia et al., suggest that CRD-guided elimination may improve the accuracy of allergen identification and reduce unnecessary food exclusions [5]. However, these findings still require validation in larger, prospective trials [4].

Beyond histologic remission, patient-centered outcomes such as symptom improvement, treatment adherence, and quality of life (QoL) have gained increasing recognition as measures of therapeutic success. Braseth et al. observed symptomatic relief in a substantial proportion of adults on test-guided diets despite modest histologic remission rates, underscoring the multifactorial nature of therapeutic response in EoE [6]. Simplified dietary approaches, including step-up or single-food elimination protocols, may enhance long-term adherence without compromising efficacy, particularly in pediatric populations [8,9].

Integrating molecular diagnostics with individualized nutrition frameworks aligns with the broader paradigm of precision nutrition, aiming to optimize dietary safety, efficacy, and patient quality of life. Therefore, this review summarizes current evidence on elimination-based therapies in EoE and evaluates the evolving role of molecular allergology—particularly component-resolved diagnostics (CRD)—in the transition from empiric to precision-guided management.

Objectives of the Review

The objective of this narrative review is to critically examine the role of component-resolved diagnostics (CRD) and related molecular tools in the diagnosis and management of eosinophilic esophagitis (EoE). Specifically, we aim to (i) evaluate current evidence on the diagnostic accuracy and clinical applicability of CRD compared with conventional allergy testing, (ii) explore how CRD-guided and molecularly informed dietary strategies contribute to precision nutrition in EoE, and (iii) identify existing limitations and future research directions required to translate molecular diagnostics into validated, patient-centered clinical practice.

Search Strategy and Selection Criteria

This narrative review is based on a structured literature search conducted in the PubMed and Scopus databases covering the past 25 years, using combinations of search terms including “eosinophilic esophagitis,” “component-resolved diagnostics,” “molecular allergy,” “food allergy,” and “elimination diet.” The selection included original research articles, clinical trials, meta-analyses, and comprehensive reviews published in English. Data extraction and synthesis were performed independently by the authors, with priority given to studies demonstrating methodological rigor, clinical relevance, and a direct contribution to the understanding of diagnostic and nutritional strategies in EoE. The most representative and methodologically robust studies were summarized in the summary tables to provide a concise overview of current evidence.

A proposed diagnostic workflow is illustrated in Figure 1.

Schematic representation of the diagnostic pathway in eosinophilic esophagitis (EoE). Patients typically present with symptoms of esophageal dysfunction, most commonly dysphagia and food impaction. Endoscopy with biopsy remains the diagnostic cornerstone, requiring histologic confirmation of ≥15 eosinophils per high-power field. The workflow includes the exclusion of alternative causes of esophageal eosinophilia (e.g., gastroesophageal reflux disease, infections, connective tissue disorders). Molecular diagnostics-such as serum food-specific IgE, skin prick testing (SPT), atopy patch testing (APT), and CRD (multiplex microarrays including Immuno Solid-phase Allergen Chip; ISAC or Allergy Explorer 2; ALEX2)-serve as complementary tools for identifying potential food triggers. Integration of clinical, endoscopic, histologic, and molecular data enables phenotype stratification and informs personalized dietary or pharmacologic strategies. Created by authors in Microsoft PowerPoint.

## 2. Treatment of Eosinophilic Esophagitis

Dietary intervention remains the cornerstone of EoE management, as it directly targets the underlying immune response to food antigens. Across randomized controlled trials, cohort studies, and meta-analyses, a consistent gradient of efficacy has been observed: elemental diets yield the highest histologic remission rates, followed by empiric multi-food elimination, whereas allergy test-based or molecularly guided strategies show more variable outcomes [1,4,10].

Step-up and milk-only exclusion approaches have emerged as simplified alternatives that preserve reasonable efficacy while improving adherence, especially in pediatric cohorts [1]. Although molecularly or serologically guided interventions (e.g., based on IgE, IgG4, or CRD) offer the promise of personalization, current evidence suggests that empiric and elemental diets remain the most reliable therapeutic standards. As molecular diagnostics continue to evolve, their integration with diet-based management may facilitate a transition toward precision nutrition in EoE.

### 2.1. Standard Dietary Comparators

Dietary therapy remains a central component of EoE management, directly addressing the immune-mediated reaction to food antigens. Across randomized controlled trials, cohort studies, and meta-analyses, a reproducible hierarchy of efficacy has been identified: elemental diets yield the highest histologic remission rates, followed by empiric multi-food elimination, whereas allergy test–based or molecularly guided strategies show more variable results [4,11,12,13].

Elemental diets achieve remission in more than 90% of patients, supported by strong evidence from both pediatric and adult cohorts. Despite their efficacy, their use is limited by challenges in palatability, social acceptability, and long-term adherence [14,15]. Empiric elimination approaches, particularly the six-food elimination diet (SFED), have demonstrated robust effectiveness, with remission rates ranging from 50% to 80% across multiple prospective studies and meta-analyses [16,17,18]. Stepwise simplifications, including four-food (4FED) and two-food (2FED) versions, maintain satisfactory efficacy while improving adherence and nutritional sustainability [19,20].

Recent studies have evaluated “step-up” frameworks in which dietary restriction intensity escalates only when initial limited elimination fails to induce remission. This approach minimizes unnecessary exclusions and has shown cumulative remission rates up to 79%, while enhancing patient quality of life [10]. Similarly, single-food or milk-only exclusion diets have gained popularity, particularly in pediatric populations where cow’s milk represents the predominant antigenic trigger [14,21]. These simplified protocols balance effectiveness and feasibility, achieving remission rates of 60–77% and improved compliance compared with multi-food restrictions [2,22].

Comparative analyses reaffirm this efficacy gradient: elemental diets remain the most potent but least practical; empiric multi-food elimination offers a balance between efficacy, adherence, and quality of life; and stepwise or milk-only approaches achieve moderate yet clinically meaningful outcomes [4,15,20].

As illustrated in Table 1, elemental diets consistently produce the highest histologic remission rates (≈90–96%), empiric six- and four-food elimination diets yield intermediate efficacy (≈50–80%), while step-up and milk-only protocols demonstrate moderate but improved tolerability. Test-based or molecularly guided elimination strategies remain less effective and require further validation [23,24].

Elemental diets remain the benchmark for inducing histologic remission in EoE. Empiric six- and four-food elimination diets balance efficacy with adherence, while simplified approaches such as step-up or milk-only exclusion maintain moderate efficacy with improved tolerability.

### 2.2. IgE-Centric and Combined Allergy Testing

Allergy test–directed dietary strategies have long been explored in EoE as alternatives to empiric elimination, aiming to tailor interventions based on identifiable food sensitizations. The most commonly employed modalities include skin prick testing (SPT), atopy patch testing (APT), and serum food-specific immunoglobulin E (sIgE) assays, often used in combination to enhance diagnostic accuracy. Early pediatric studies by Spergel et al. [26] and Henderson et al. [2] demonstrated that these tests could guide partial histologic remission in a subset of patients, although overall success rates remained modest compared with empiric or elemental approaches.

Evidence from systematic reviews and meta-analyses supports a consistent pattern of efficacy. Reviews by Arias et al. (2014, 2024) [1,4] and Pitsios et al. (2022) [3] concluded that test-directed diets achieve substantially lower histologic remission rates than empiric six-food elimination or elemental diets. Pooled analyses reported average remission rates of approximately 39–66%, compared with 63–94% for empiric or elemental interventions. These findings are supported by both prospective and retrospective studies in adults and children, in which neither SPT nor sIgE reliably predicted causative foods [23,24].

The limited predictive value of allergy testing reflects the immunopathologic complexity of EoE, involving both IgE-mediated and non–IgE-mediated mechanisms. While SPT identifies immediate hypersensitivity and APT captures delayed T cell–mediated responses, their combined use (SPT + APT) increases false-positive rates and does not consistently improve diagnostic accuracy [23]. Dellon et al. [24] and Pesek et al. [27] further demonstrated that even when tissue biomarkers such as IgG4 or CD4^+^ T-cell reactivity are included, discriminatory power remains low, underscoring that sensitization profiles often exceed clinically relevant triggers.

Despite these limitations, test-based strategies retain certain clinical value. They may help guide focused elimination in polysensitized or nutritionally vulnerable patients for whom broad empiric exclusion is impractical. Moreover, the ongoing development of molecular assays—integrating CRD and multiplex platforms—suggests potential for future refinement beyond conventional skin or serum testing.

Overall, evidence accumulated over three decades indicates that SPT-, APT-, and sIgE-guided diets are inferior to empiric and elemental interventions in achieving histologic remission but may serve as adjuncts to emerging molecular diagnostics within a precision-nutrition framework for EoE management [28]. Their inferiority reflects not only lower therapeutic efficacy but also the inconsistent predictive accuracy of these tests in identifying true causal allergens. As summarized in Table 2, outcomes of IgE- and test-directed elimination diets vary considerably across studies. Early findings suggested partial benefits from SPT- and APT-guided approaches, but subsequent analyses consistently demonstrated lower remission rates and predictive accuracy compared with empiric or elemental diets. Altogether, conventional IgE-based testing lacks sufficient precision to serve as a standalone guide for dietary elimination. Building upon these limitations, component-resolved and multiplex molecular diagnostics have emerged as next-generation tools aimed at refining dietary personalization in EoE.

### 2.3. Age and Geographic Differences

Age and geography contribute meaningfully to variability in dietary response and clinical presentation. Although elimination diets are broadly effective across age groups, consistent differences in disease behavior, allergen patterns, and adherence have been reported [25].

Adult cohorts dominate the literature, whereas pediatric data remain more limited and geographically restricted. In a U.S. retrospective study, Wolf et al. [25] reported approximately 39% histologic remission after targeted or six-food elimination, accompanied by substantial symptomatic and endoscopic improvement. Conversely, pediatric cohorts—such as the prospective study by Pesek et al. [27]—achieved higher remission (~59%) with allergen-specific or elemental approaches, suggesting greater immune plasticity early in life. The step-up 2-4-6 model evaluated by Molina-Infante et al. [20] indicated that age did not significantly alter histologic outcomes, although pediatric patients tended to show faster symptom relief.

Systematic reviews generally confirm comparable efficacy in both adults and children but emphasize the scarcity of age-stratified evidence. Mayerhofer et al. [12] found similar remission rates across mixed-age cohorts, while Arias et al. [1] noted that elemental and empiric strategies outperform test-directed diets regardless of age. In contrast, Nuyttens et al. [9] observed that less restrictive one- or two-food eliminations in children were associated with improved quality of life, underscoring feasibility as a key determinant of long-term success. Taken together, age appears to influence adherence and quality-of-life outcomes more than histologic efficacy itself.

Geographic variability adds further complexity. Most evidence originates from North America and Western Europe, with limited data from regions characterized by distinct dietary habits and allergen exposures. Only isolated reports (e.g., Arias et al. [1]) have addressed regional adaptation of food lists, while environmental factors such as mold or cockroach sensitization have been linked to reduced response in U.S. pediatric cohorts [2]. Overall, geographically diverse, age-stratified studies are needed to refine dietary recommendations and improve global applicability.

### 2.4. Step-Up and Step-Down Dietary Frameworks

Recent approaches increasingly prioritize balancing efficacy with quality of life. Step-up strategies begin with the elimination of one or two foods and expand only if remission is not achieved, aiming to minimize unnecessary restriction and invasive procedures. Step-down strategies start broadly (e.g., six-food elimination) and progressively liberalize the diet after remission, allowing systematic food reintroduction [19,20].

Across multicenter and prospective studies, step-up frameworks have shown promising remission rates while improving adherence and reducing the need for repeated endoscopies. Molina-Infante et al. [19] first demonstrated 54% remission with a fixed four-food elimination, increasing to 72% with step-up rescue. A subsequent multicenter study involving 130 patients (including 25 pediatric cases) confirmed progressive efficacy across two-, four-, and six-food sequences—43%, 60%, and 79%, respectively [20]. In an open-label RCT, Kliewer et al. [10] found that one-food (milk-only) elimination induced remission in one-third of adults, while escalation to six-food improved outcomes without excessive burden. Meta-analytic data indicate that step-up frameworks achieve a mean remission rate of approximately 60%, with most patients reacting to only one or two triggers [4]. The comparative characteristics of step-up and step-down elimination strategies are summarized in Table 3.

Building on these adaptive models, research has explored combining pharmacologic agents—topical corticosteroids, proton pump inhibitors (PPIs), and emerging biologics—with elimination diets to enhance remission durability and quality of life while reducing dietary burden. From a nutritional standpoint, step-up frameworks minimize unnecessary time on highly restrictive regimens and lower the risk of nutrient deficiencies. Embedding predefined “nutrition checkpoints” (e.g., at 4–6 weeks of each step) with dietitian review and targeted laboratory monitoring can prevent cumulative deficits while maintaining therapeutic momentum [19].

### 2.5. Combined Modalities and Adjunctive Pharmacotherapy

A multimodal approach that integrates dietary and pharmacologic interventions is becoming increasingly relevant in EoE. While elimination diets target antigen-driven inflammation, pharmacotherapy—including topical corticosteroids, PPIs, and biologics—addresses overlapping inflammatory pathways to promote mucosal healing and symptom control. This rationale reflects the heterogeneity of the disease and the recognition that no single therapy ensures durable remission across all phenotypes [29,30,31].

Systematic reviews and cohort studies indicate that combination therapy yields superior outcomes compared with monotherapy. Dutta et al. [29] reported that diets combined with PPIs or topical steroids improved remission in refractory EoE, with histologic response rates ranging from 54% for six-food elimination to 87% for one-food elimination when paired with PPI therapy. In a pediatric cohort, Constantine et al. [30] found that topical steroids plus test-based elimination achieved a 91% clinical response—significantly higher than steroids (71%) or diet (64%) alone. Similarly, Reed et al. [31] observed a reduction in median eosinophil counts from 51 to 2 eos/hpf, along with parallel improvements in dysphagia when topical steroids were added to a two-food elimination. These findings support the use of pharmacologic immunomodulation to potentiate dietary efficacy.

Biologic therapies provide an additional adjunct for refractory or severe disease. Dupilumab, an IL-4 receptor-α antagonist targeting Th2 pathways, has shown robust reductions in histologic eosinophilia and symptom burden. Spergel et al. [32] reported that 22 of 26 patients achieved ≤6 eos/hpf, with 24 experiencing complete symptom resolution during therapy. Broader reviews (Lucendo and Molina-Infante [33]; Nhu and Aceves [34]) describe histologic remission rates of approximately 60–65%, with significant improvements in dysphagia scores and reduced steroid dependence. Early data for cendakimab and lirentelimab further expand the therapeutic horizon, although long-term comparative trials remain limited.

The safety profiles of combination strategies are generally acceptable. Adverse events include mild esophageal candidiasis (≈5–30%) and transient adrenal suppression associated with topical steroids [35,36]. Dupilumab-related events—primarily injection-site reactions and conjunctivitis—are typically infrequent and manageable [37]. The main non-pharmacologic concern remains the risk of nutritional deficiencies and psychosocial stress from highly restrictive diets, particularly in children [38].

Overall, combining dietary and pharmacologic modalities offers complementary mechanisms that can achieve deeper and more durable remission while improving quality of life. As biologists enter routine clinical practice, EoE management is expected to evolve toward a precision-guided, multimodal framework integrating molecular diagnostics, targeted pharmacotherapy, and personalized nutrition.

### 2.6. Summary and Transition to Molecular Approaches

Empiric elimination diets remain the most effective therapeutic approach for eosinophilic esophagitis (EoE), achieving histologic remission in approximately 60–95% of cases depending on the number of eliminated foods and patient age [1,4,12,14,15,16,17,18,19,20]. Step-up and step-down strategies have improved practicality, allowing clinicians to balance efficacy, adherence, and nutritional adequacy [10,19,20]. Nevertheless, empiric frameworks are limited by their trial-and-error nature, the need for multiple endoscopies, and potential nutrient deficiencies—particularly in pediatric populations, where restrictive diets may compromise growth and micronutrient status [1,2,39,40,41]. Despite these drawbacks, dietary therapy remains central, as no pharmacologic option currently achieves durable histologic control after discontinuation [29,35,36,37].

Traditional allergy testing—including skin prick tests (SPT), atopy patch tests (APT), and serum-specific IgE—has shown poor correlation with histologic remission and limited predictive value for identifying food triggers. Pooled remission rates range from only 39–66% for test-directed diets, compared with 63–94% for empiric or elemental approaches [1,3,4,23,27]. This gap underscores the limitations of current diagnostic tools, which detect sensitization but not necessarily clinical relevance. Similarly, pharmacologic options such as topical corticosteroids and proton pump inhibitors (PPIs) can induce remission but are associated with frequent relapses, partial symptom persistence, and adverse effects including esophageal candidiasis and adrenal suppression [31,35,36,37]. Thus, neither current dietary nor pharmacologic interventions fully address the need for durable, personalized, and nutritionally safe disease control.

These diagnostic and therapeutic limitations highlight the need for precision-based approaches that integrate immunologic mechanisms with individualized nutrition. Molecular allergology and component-resolved diagnostics (CRD) have emerged as innovative tools that identify sensitization at the level of specific allergenic proteins rather than crude extracts. By differentiating genuine sensitization from cross-reactivity, CRD may refine elimination strategies, reduce unnecessary dietary restrictions, and bridge the gap between diagnostic precision and nutritional safety.

The following section explores the diagnostic accuracy, clinical utility, and translational potential of CRD and related molecular methods in EoE. By aligning diagnostic precision with individualized nutritional strategies, these molecular tools may represent a paradigm shift in EoE management—a focus examined in detail in the next section.

## 3. Component-Resolved Diagnostics and Molecular Approaches in Eosinophilic Esophagitis

### 3.1. Diagnostic Accuracy and Clinical Relevance of CRD

Building on the limitations of empiric and test-directed approaches outlined above, molecular allergology has emerged as a promising frontier in EoE diagnostics. Component-resolved diagnostics (CRD) and multiplex allergen platforms represent key innovations for individualizing dietary management in EoE. Unlike conventional methods such as skin prick testing (SPT) or serum food-specific immunoglobulin E (sIgE), CRD identifies sensitization to individual allergenic molecules rather than whole extracts, enabling differentiation between true primary sensitization and cross-reactivity. Integrating these molecular insights into dietary decision-making aims to reduce unnecessary food exclusions while maintaining histologic efficacy [5,24].

The diagnostic accuracy of allergy testing in EoE remains a major challenge when translating immunologic sensitization into clinically actionable dietary guidance. Across available studies, CRD demonstrates superior detection of allergen sensitization compared with traditional methods, offering greater precision in identifying clinically relevant food triggers. In a prospective study, Armentia et al. [5] found that CRD identified specific sensitizations in 87.6% of patients with EoE, and subsequent CRD-guided interventions produced marked eosinophil reduction and symptom improvement in 75.2% of cases. These data suggest that molecular allergen profiling can delineate disease-relevant immune pathways, particularly for cross-reactive plant proteins such as profilins and PR-10, which are frequently implicated in adult-onset EoE [42].

In adults, Dellon et al. [24] applied a T-cell–based assay and tissue IgG4 profiling for five major food groups. Although histologic remission was achieved in only about one-fifth of cases, immunologic readouts showed moderate concordance (53–75%) between identified allergens and known triggers, indicating partial diagnostic accuracy. Lim et al. [43] expanded this concept in a prospective controlled trial using serum food-specific IgG4 (milk, wheat, soy, egg, nuts), observing histologic remission in nearly half of participants with comparable clinical improvement. Despite methodological diversity, a recurring pattern emerges: molecular and serologic assays capture disease-relevant immune activity but remain imperfect surrogates for dietary responsiveness.

By contrast, conventional tests—including SPT, APT, and serum-specific IgE—show substantial variability in predictive performance. Large retrospective cohorts report low positive predictive values (<50%) for SPT and APT, whereas negative predictive values are generally higher (up to 92%) [3,44]. Sensitivity for sIgE-based testing may approach 87.5%, but specificity remains limited (~68%), indicating that positive results often reflect sensitization without clinical reactivity [23]. Moreover, APT alone exhibits poor diagnostic yield, with sensitivity as low as 5.9% [45]. Meta-analyses reinforce these limitations, reporting tissue-level improvement in only 39–53% of patients on test-directed diets compared with 63–79% under empiric elimination [3,46]. Emerging immune assays—such as T-cell proliferation or tissue IgG4 profiling—provide mechanistic insights but currently lack validation in large prospective cohorts [24].

Comparative evaluations further support this trend. Whereas empiric six-food or elemental diets achieve remission in >70–90% of cases, test-directed and CRD-guided strategies typically yield 40–65% [1,4]. Arias et al. [47] and De Vlieger et al. [48] reported that CRD-enhanced approaches rarely surpass empiric elimination in inducing histologic remission, though they may improve adherence by focusing on fewer foods. Álvarez Hodel [49] observed higher remission in CRD-negative individuals, suggesting that the absence of molecular sensitization could predict favorable outcomes under less restrictive regimens.

Taken together, CRD emerges as a promising adjunct for refining dietary therapy in EoE. Nevertheless, its clinical use remains largely investigational, and direct comparative trials are needed to establish its predictive value against empiric or conventional test-directed strategies [50]. CRD and multiplex diagnostics constitute an important step toward precision nutrition in EoE, adding molecular granularity that complements empiric frameworks and supports more nuanced decisions for patients at nutritional risk or with poor adherence to broad eliminations. As technology evolves, integrating CRD into personalized algorithms—potentially alongside biologics or targeted pharmacotherapy—may shift EoE management from empiric exclusion toward molecularly guided precision care [50].

### 3.2. CRD-Guided Elimination Diet Strategies

Component-resolved diagnostics (CRD) have gained attention as a tool to personalize elimination diets in eosinophilic esophagitis (EoE), aiming to achieve a better balance between therapeutic efficacy and dietary restriction. By mapping sensitization to individual allergenic components—such as lipid transfer proteins (LTPs) and PR-10 proteins—CRD enables a more targeted approach than empiric or conventional test-based methods.

In an observational cohort study, Armentia et al. [5] used molecular microarrays in 87 patients with EoE and identified frequent sensitizations to grass group 1 allergens and lipid transfer proteins from peach (Pru p 3) and mugwort (Art v 3). A CRD-guided elimination combined with allergen immunotherapy produced clinical improvement in 78.3% of participants and induced histologic normalization in approximately three-quarters of cases, indicating that molecular profiling may inform both dietary and immunologic management strategies [5].

In parallel, Dellon et al. [24] piloted an immune signature–based approach integrating CD4^+^ T-cell proliferation and tissue IgG4 quantification. Although tissue-level improvement was observed in only 21% of cases, symptom relief reached 68%, and the mean number of excluded foods was limited to 3.4—suggesting that immunologic precision may reduce dietary burden even when overall remission remains modest.

Comparative evaluations with traditional testing reinforce this view. Whereas empiric six-food or elemental diets achieve remission in >70–90% of cases, test-directed and CRD-guided strategies typically yield 40–65% [1,4]. Arias et al. [1,4] and De Vlieger et al. [48] demonstrated that CRD-enhanced approaches rarely surpass empiric elimination in inducing histologic remission, though they may improve adherence by focusing on fewer foods. Álvarez Hodel [49] reported higher remission rates in CRD-negative individuals, suggesting that the absence of molecular sensitization could predict favorable outcomes under less restrictive regimens.

A comparative summary of remission rates and diagnostic accuracy across CRD-guided, empiric, and elemental diet approaches is presented in Table 4, illustrating the relative clinical performance and methodological diversity of available studies.

From both nutritional and clinical perspectives, CRD-guided elimination can individualize dietary plans and identify non-food sensitizations relevant to EoE pathogenesis. However, current data remain limited by small sample sizes, a lack of head-to-head comparisons, and inconsistent histologic reporting. Larger prospective studies are required to determine whether CRD can outperform—or reliably complement—empiric approaches while minimizing dietary restriction. Integrating CRD results into structured, dietitian-supervised care could help ensure that immunologic precision translates into nutritionally safe and sustainable elimination protocols, thereby reducing long-term risks of deficiency and improving adherence.

### 3.3. Clinical Advantages and Limitations of CRD

Component-resolved diagnostics (CRDs) provide several advantages for enhancing diagnostic precision and supporting individualized management in EoE. In contrast to extract-based assays, CRD measures IgE reactivity to purified or recombinant allergen molecules, enabling detailed mapping of cross-reactivity and co-sensitization patterns. Findings from prospective and cohort studies consistently report polysensitization rates above 80% and reveal cross-reactive epitopes—particularly lipid transfer proteins (LTPs) and birch-related profilins—frequently implicated in adult-onset EoE [5,51,52]. Such molecular insights can guide targeted elimination strategies and, in selected cases, inform the use of adjunctive allergen immunotherapy.

The clinical benefits of CRD-guided care are best illustrated by Armentia et al. [5], in which molecular profiling enabled precise identification of relevant food and aeroallergen triggers. Their combined CRD-guided elimination and immunotherapy protocol produced sustained clinical improvement and histologic normalization in approximately three-quarters of participants, with durable disease control maintained over three years. van Rhijn et al. [52] further demonstrated that molecular chips (e.g., ISAC) can map complex sensitization networks, including pollen–food cross-reactivity, thereby supporting targeted rather than broad empiric restriction. Similar patterns have been observed in broader allergic diseases, where clinical cross-reactivity syndromes link pollen allergens with homologous plant-derived food proteins, such as profilins, PR-10, and lipid transfer proteins (LTPs). Additionally, structurally conserved panallergens—including tropomyosins, parvalbumins, and caseins—contribute to cross-reactivity between aeroallergens, plant-derived foods, and animal proteins. These molecular interactions underscore the clinical relevance of CRD in distinguishing true food allergy from pollen-associated or panallergen-mediated sensitization in EoE and related disorders [53,54].

From a nutritional standpoint, CRD-guided elimination may enhance dietary safety by minimizing unnecessary exclusions while maintaining macro- and micronutrient adequacy. Structured dietitian oversight remains essential to ensure that the molecular precision achieved in diagnostics translates into nutritionally sustainable interventions [54,55].

Despite these mechanistic advantages, several practical barriers limit routine clinical use. High costs, restricted availability, and the need for specialized interpretation constrain broader implementation [56]. Testing is time-consuming, and results require careful integration with clinical history, dietary feasibility, and patient context. These factors contribute to the limited adoption of CRD in daily practice, where empiric dietary frameworks remain predominant.

Furthermore, multiple methodological and biological limitations hinder CRD’s clinical translation. Correlations between molecular sensitization profiles and conventional IgE assays (e.g., ImmunoCAP) are often weak, reflecting the multifactorial and tissue-specific nature of EoE rather than a systemic IgE-driven allergy [24,43]. Emerging evidence also suggests that competing antibody subclasses—particularly IgG4—may interfere with allergen recognition and modulate assay results, complicating interpretation and potentially masking clinically relevant sensitizations. In addition, response rates to CRD-guided elimination remain modest in adult populations, with histologic remission frequently below 50% even in well-defined cohorts [24,43,48]. These limitations indicate that, while CRD provides valuable molecular granularity, its diagnostic reproducibility and predictive validity still require rigorous validation before it can be routinely applied in clinical practice.

Methodological issues—including small sample sizes, short follow-up durations, and a lack of standardized histologic endpoints—further limit the strength of current evidence [57]. Sensitivity and specificity vary widely across assay platforms, and no randomized controlled trials have yet compared CRD-guided versus empiric elimination strategies in a head-to-head design. Moreover, assay heterogeneity and the absence of unified interpretative criteria continue to hinder reproducibility across centers [58].

Beyond IgE-centric testing, Min et al. [55] highlighted the potential of transcriptomic panels such as the Eosinophilic Esophagitis Diagnostic Panel (EDP), which achieved 85% sensitivity and correlated strongly with histologic inflammation (r = −0.73), illustrating how molecular biomarkers may complement CRD in future integrated diagnostic algorithms.

Consequently, while CRD offers unique molecular resolution, its limited validation and infrequent use in daily clinical settings currently preclude it from serving as a standalone diagnostic or therapeutic tool. Future studies should aim to integrate CRD with molecular, clinical, and nutritional parameters to develop a validated framework for precision-based dietary interventions in EoE.

### 3.4. Future Directions and Research Gaps for CRD in EoE

The future of EoE diagnostics lies in integrating component-resolved diagnostics (CRD) with multimodal molecular and immunologic approaches. Current evidence underscores the need for combined strategies that unite CRD with functional T-cell or cytokine assays and robust clinical phenotyping [58]. Such integration may overcome the limited sensitivity and predictive power of single-platform testing, enabling endotype-specific approaches that link molecular sensitization to immune activation and dietary response.

Although CRD demonstrates high allergen detection rates in some cohorts (up to 87.6%), study heterogeneity and limited validation remain major challenges [5]. Advances in transcriptomics and genomics extend diagnostic capabilities beyond sensitization alone. The EoE Diagnostic Panel (EDP)—a 96-gene qPCR assay—has reported 96% sensitivity and 98% specificity, accurately distinguishing EoE from controls and predicting relapse [59,60]. Such molecular signatures may support dynamic disease monitoring and the identification of endotype-specific therapeutic targets.

Epigenetic mechanisms also play a regulatory role in EoE pathogenesis and treatment response. Lu et al. identified dysregulation of miR-21, miR-223, and miR-375, which normalized following corticosteroid therapy, supporting their utility as dynamic disease biomarkers [61]. Jensen et al. [62] reported altered DNA methylation at UNC5B and ITGA6 genes predictive of corticosteroid responsiveness. Similarly, Markey et al. [63] demonstrated that miR-155 overexpression disrupts epithelial barrier integrity by suppressing CLDN7, linking epigenetic control to barrier dysfunction. Collectively, these findings suggest that miRNA expression and DNA methylation patterns could serve as non-invasive biomarkers of disease activity and therapeutic response.

Additional diagnostic layers extend beyond genomics and proteomics. EoE pathogenesis involves not only Th2-driven cytokine signaling and epithelial barrier impairment but also complex interactions among epithelial cells, eosinophils, and fibroblasts that perpetuate tissue remodeling [64]. Extracellular vesicles (EVs), which transport miRNAs, proteins, and lipids, have gained particular attention as mediators of intercellular communication within the esophageal microenvironment. Circulating EV-derived miRNAs have been proposed as minimally invasive biomarkers that mirror both local inflammation and systemic immune activation [65]. Integrating EV and miRNA signatures into multi-omics frameworks could improve diagnostic specificity, facilitate patient stratification, and enable molecular monitoring in EoE.

Future CRD-integrated algorithms should incorporate nutrition-focused endpoints and biomarkers (e.g., 25-hydroxyvitamin D, ferritin, transferrin saturation, vitamin B_12_, folate, zinc, iodine status) alongside validated patient-reported outcomes addressing food-related quality of life. Parallel evaluation of the gut microbiome and fermentation-derived metabolites (e.g., short-chain fatty acids) may help predict tolerance during food reintroduction and identify patients at risk of dysbiosis in restrictive phases [1,2,3].

Recent reviews highlight that genomics, proteomics, metabolomics, and epigenetics can identify key pathogenic pathways and biomarkers—including eotaxin-3, interleukin-13, and galectin-3—supporting a shift toward mechanistic subtyping [66,67,68]. However, integration remains challenging due to methodological heterogeneity and the absence of standardized bioinformatic pipelines. Clinical translation will require multicenter validation, cost-effectiveness analyses, and user-friendly diagnostic workflows.

The convergence of molecular diagnostics with therapeutic innovation represents a major opportunity. CRD-derived sensitization profiles may inform biologic selection or guide adjunctive therapies targeting IL-5, IL-13, or TSLP pathways. Integrating CRD with biologics or targeted pharmacotherapy [50] could transform EoE management into a fully personalized framework uniting molecular, immunologic, and nutritional precision. Ultimately, progress will depend on bridging experimental omics data with practical clinical tools to achieve a validated, cost-effective, and patient-centered model of care.

Multi-omics research continues to unravel the complex molecular architecture of EoE, revealing interconnected layers of immune, epithelial, and metabolic dysregulation. Transcriptomic profiling remains the most extensively applied approach, expanding the canonical EoE transcriptome to over 1600 dysregulated genes and novel long non-coding RNAs such as BANCR, which modulates IL-13–induced inflammatory signaling [69]. Epigenomic analyses focusing on microRNA expression (e.g., miR-21, miR-223, miR-375) demonstrate reversible regulation during glucocorticoid-induced remission, underscoring their value as biomarkers of treatment response [61]. Proteomic and metabolomic integration has identified distinct immune and metabolic signatures in both plasma and tissue, including altered vitamin B6 and amino acid pathways, suggesting that systemic metabolic changes mirror local inflammation [70]. Microbiomic and metatranscriptomic findings indicate reduced microbial diversity and enrichment of pro-inflammatory taxa, reinforcing the hypothesis that host–microbe interactions influence mucosal immunity and epithelial barrier integrity [71].

Collectively, these studies demonstrate that multi-omics approaches enable cross-compartmental integration of local and systemic signatures, supporting biomarker discovery and mechanistic insight. Despite encouraging diagnostic accuracy in discovery cohorts (AUC up to 0.93; sensitivity 87.5%; specificity 93.3%), most findings remain constrained by small sample sizes, methodological heterogeneity, and limited external validation. Future research integrating genomics, proteomics, metabolomics, microbiomics, and epigenomics with dietary and clinical data may enable endotype-based classification and facilitate precision nutrition strategies in EoE.

The integration of CRD and multi-omics approaches offers unprecedented insight into the immunologic and molecular heterogeneity of EoE. However, translating this complexity into patient-centered dietary management requires structured nutritional frameworks that ensure both efficacy and safety. The following section focuses on the practical implementation of precision nutrition in EoE, emphasizing dietitian-led care, nutritional adequacy, and shared decision-making as essential components for transforming molecular precision into sustainable clinical benefit.

### 3.5. Nutrition Considerations and Practical Implementation

The translation of molecular insights into clinical care requires a structured nutritional framework that bridges diagnostic precision with real-world dietary feasibility. While CRD and multi-omics tools can identify relevant sensitizations and immune pathways, their integration into elimination diet strategies must be guided by principles of nutritional safety. Immunologic precision does not automatically ensure dietary adequacy; therefore, CRD-guided elimination should always be implemented under the supervision of experienced dietitians to balance therapeutic efficacy with long-term nutritional sustainability.

Dietary therapy in EoE requires structured nutrition care to balance anti-inflammatory efficacy with the maintenance of nutritional adequacy and adherence. Nutritional management must therefore be individualized and dietitian-supervised to prevent iatrogenic deficiencies and ensure long-term sustainability [39,40,41]. Baseline assessment should include a detailed dietary history, anthropometric evaluation, and targeted biochemical testing—particularly for 25(OH)D, iron indices, vitamin B_12_, folate, and zinc. In pediatric cohorts, growth velocity and mid–upper arm circumference are essential parameters for monitoring nutritional status [39,40].

Elimination phases should be accompanied by tailored substitution plans to maintain nutrient balance and preserve quality of life. Fortified dairy alternatives can mitigate calcium, iodine, and vitamin D deficiencies, while high-quality plant proteins may compensate for the exclusion of milk or soy as primary protein sources [65,66]. Whole-grain, gluten-free products enriched with B vitamins and fiber can reduce the nutritional impact of wheat avoidance. Texture modification and behavioral strategies—such as slow eating, mastication training, and liquid “washdown” techniques—may also help overcome dysphagia-related caloric restriction [41].

Nutrient deficiencies and psychosocial burden remain major limitations of long-term elimination diets. Observational studies report suboptimal intakes of calcium, vitamin D, iron, and B vitamins—particularly in children and adolescents undergoing multi-food restriction [1,2]. Early dietetic intervention, individualized supplementation, and progressive food reintroduction are therefore critical for maintaining nutritional adequacy. Incorporating the gut microbiome into nutritional frameworks may further enhance tolerance to food reintroduction and prevent dysbiosis associated with prolonged dietary restriction [72].

Finally, a shared decision-making model that integrates CRD results with patient preferences, food culture, and affordability is recommended to enhance adherence and optimize patient-centered outcomes. Dietitian-led care pathways—including regular nutritional assessments, biomarker monitoring, and education on safe reintroduction protocols—are key to the successful implementation of precision nutrition in EoE [39,40,41]. Ultimately, embedding CRD- and omics-guided insights within multidisciplinary, dietitian-led frameworks may ensure that molecular precision translates into nutritionally safe, sustainable, and patient-centered management of EoE.

To synthesize the diagnostic and nutritional dimensions discussed above, Figure 2 presents an integrated overview of component-resolved diagnostics (CRD) and molecular approaches within the framework of precision nutrition in eosinophilic esophagitis (EoE).

Figure 2 Integrated diagnostic and nutritional framework for eosinophilic esophagitis (EoE) based on component-resolved diagnostics (CRD) and molecular approaches. The diagram summarizes the transition from conventional extract-based tests toward molecular allergen profiling and multi-omics integration. CRD identifies specific allergen components (e.g., LTP, PR-10, profilins, tropomyosins, caseins), which can be combined with immunologic assays (IgE/IgG4, T-cell proliferation, cytokine mapping) to guide individualized elimination diets. Dietitian-supervised implementation, nutritional monitoring (25(OH)D, ferritin, vitamin B_12_, folate, zinc), and microbiome support ensure that molecular precision translates into nutritionally safe and sustainable management—advancing EoE care toward precision nutrition.

## 4. Conclusions

Component-resolved diagnostics (CRDs) represent an emerging precision tool in the management of eosinophilic esophagitis (EoE), linking molecular sensitization profiles to individualized dietary therapy. By identifying IgE reactivity to specific allergen components rather than whole extracts, CRD provides greater diagnostic precision than conventional extract-based tests, potentially reducing unnecessary food exclusions and improving patient adherence.

Nevertheless, current evidence remains preliminary. Limited validation, small and heterogeneous cohorts, and variability in assay platforms constrain reproducibility and generalizability. Although some studies demonstrate histologic and symptomatic improvement with CRD-guided interventions, remission rates rarely exceed those achieved through empiric elimination diets. Therefore, CRD should currently be regarded as a complementary rather than a substitutive diagnostic approach.

Future research should focus on integrating CRD with transcriptomic, proteomic, and metabolomic biomarkers to enable mechanistic endotyping and targeted therapeutic strategies. Standardization, cost-effectiveness, and accessibility remain key challenges requiring multidisciplinary collaboration among allergists, gastroenterologists, and dietitians. Incorporating nutritional endpoints—such as micronutrient adequacy and patient-reported dietary quality—into CRD-guided frameworks will help ensure that molecular precision translates into safe, sustainable, and patient-centered management of EoE.

## Figures and Tables

**Figure 1 nutrients-17-03588-f001:**
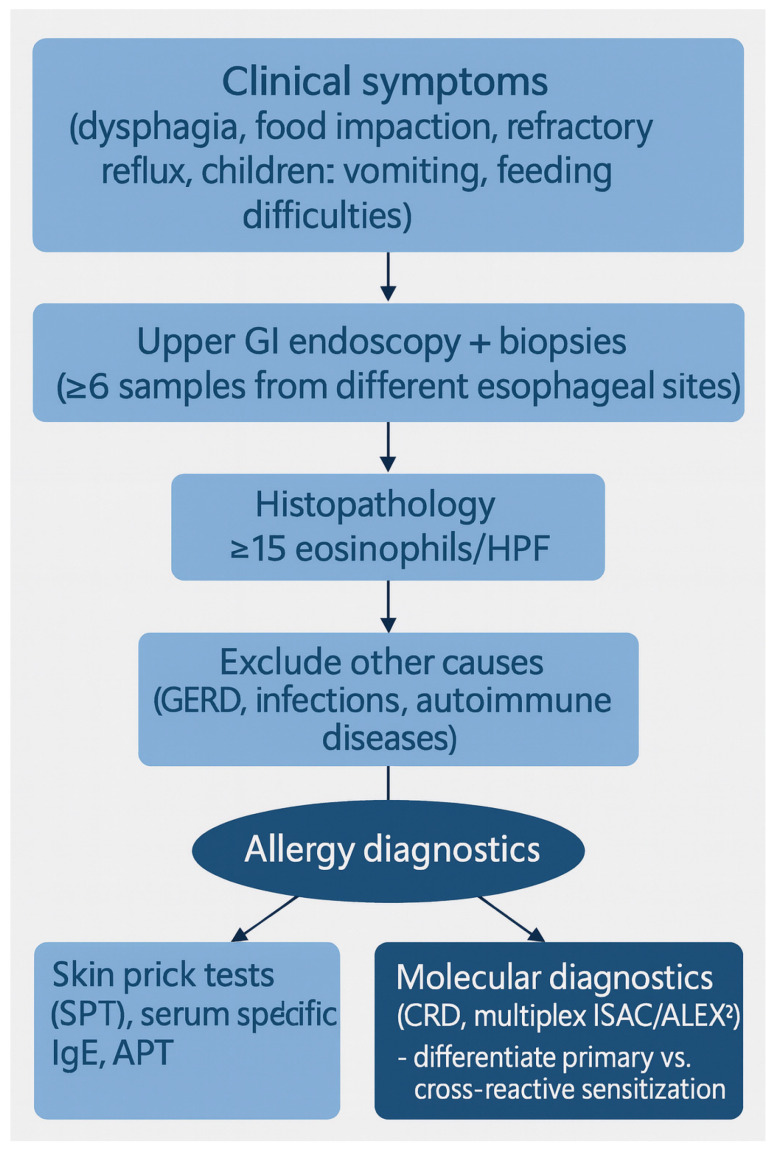
Diagnostic workflow in eosinophilic esophagitis (EoE).

**Figure 2 nutrients-17-03588-f002:**
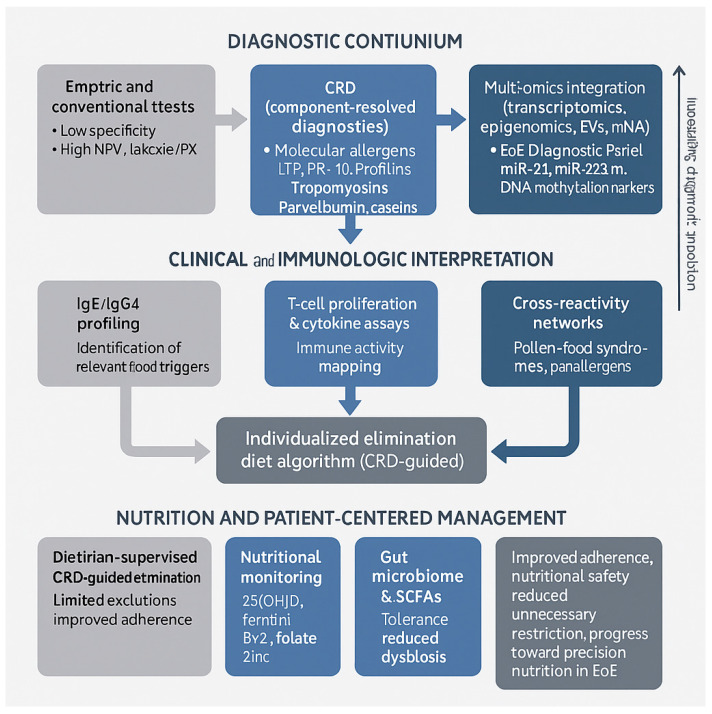
Integrated Framework of Component-Resolved Diagnostics and Precision Nutrition in Eosinophilic Esophagitis (EoE).

**Table 1 nutrients-17-03588-t001:** Efficacy of Standard Dietary Comparators in Eosinophilic Esophagitis (EoE).

Study	Population/Design	Dietary Approach	Histologic Remission (%)	Key Findings/Notes
Markowitz et al., 2003 [14]	Pediatric, prospective	Elemental diet	96	Highest efficacy; limited by adherence and palatability.
Peterson et al., 2013 [15]	Adult, prospective	Elemental diet	94	Effective in adults; confirms antigen-driven mechanism.
Arias et al., 2014 [1]	Mixed, meta-analysis	Elemental vs. empiric vs. test-based	90.8 (elemental), 72.1 (SFED), 45.5 (test-based)	Established efficacy gradient.
Arias et al., 2024 [4]	Mixed, meta-analysis	Elemental, SFED, 4FED, 2FED, 1FED	94.5, 63.9, 54.7, 44.3, 46.4 respectively	Updated pooled remission rates across diets.
Lucendo et al., 2013 [16]	Adult, prospective	SFED	73	Durable remission; supports empiric diet as first-line.
Gonsalves et al., 2012 [17]	Adult, prospective	SFED	64 (≤5 eos/hpf)	Significant symptom and histologic response.
Wolf et al., 2014 [25]	Adult, retrospective	SFED vs. targeted	56 (SFED), 32 (targeted)	Empiric diet more effective than test-based.
Kagalwalla et al., 2006 [18]	Pediatric, prospective	SFED	81	Validated efficacy in children; feasible approach.
Molina-Infante et al., 2014 [19]	Adult, multicenter, prospective	4FED	54	Simplified diet with moderate remission and improved adherence.
Molina-Infante et al., 2017 [20]	Mixed, prospective	Step-up (2 → 4 → 6 foods)	43–79 (cumulative)	Effective while reducing unnecessary restriction.
Kliewer et al., 2023 [10]	Pediatric, RCT	1-food vs. 6-food elimination	34 (1FED), 40 (SFED)	1FED non-inferior; better adherence and tolerability.
Wechsler et al., 2021 [21]	Pediatric, prospective	1-food (milk-only)	64	Supports milk-driven phenotype; high compliance.
Kagalwalla et al., 2012 [11]	Pediatric, prospective	Milk elimination	77	Confirms milk as predominant trigger in EoE.
Zalewski et al., 2022 [22]	Adult, retrospective	SFED, long-term follow-up	54–58	Sustained remission; defines chronic dietary outcomes.
Henderson et al., 2012 [2]	Pediatric, retrospective	Elemental, SFED, test-based	96, 81, 65	Elemental > empiric > test-based; consistent hierarchy.
Cotton et al., 2017 [13]	Mixed, meta-regression	SFED ± steroids	72 (diet), 84 (combination)	Combined approach yields higher remission.
Mayerhofer et al., 2023 [12]	Mixed, meta-analysis	Elemental vs. empiric vs. targeted	90 vs. 70 vs. 45	Confirms elemental superiority; empiric remains standard.

**Table 2 nutrients-17-03588-t002:** Outcomes of IgE- and Test-Directed Elimination Diets in Eosinophilic Esophagitis (EoE).

Study	Population	Testing Method(s)	Dietary Strategy	Histologic Remission (%)	Key Findings
Henderson et al., 2012 [2]	Pediatric	SPT, APT	Test-directed vs. SFED	65 (test) vs. 81 (SFED)	Empiric superior to test-directed elimination.
Spergel et al., 2012 [26]	Pediatric	SPT, APT	Combined testing-guided	53–77	Partial remission; high variability.
Rodríguez-Sánchez et al., 2014 [23]	Adult	sIgE, SPT, APT	sIgE-directed vs. SFED	73 vs. 53	No significant difference (*p* = 0.17).
Dellon et al., 2019 [24]	Adult	SPT, IgG4, CD4^+^ T-cell	Test-directed	<50	Poor predictive value; limited accuracy.
Pesek et al., 2017 [27]	Pediatric	SPT, sIgE	Test-directed	–	Low predictive value; poor correlation with triggers.
Arias et al., 2014 [1]	Mixed	SPT, sIgE	Test-directed vs. SFED	45.5 vs. 72.1	Meta-analysis; empiric superior.
Pitsios et al., 2022 [3]	Mixed	SPT, APT, sIgE	Test-directed vs. empiric	39–66	Systematic review; low predictive accuracy.
Arias et al., 2024 [4]	Mixed	SPT, sIgE	Test-directed vs. empiric	39.5 vs. 63.9	Large meta-analysis; empiric consistently superior.

**Table 3 nutrients-17-03588-t003:** Step-up versus step-down elimination strategies.

Framework	Approach	Example Protocol	Strengths	Limitations	Best Suited For	Key References
Step-down	Begin broad, then liberalize	Start with SFED → sequential reintroduction	High initial remission; systematic	Burdensome initially; risk of over-restriction	Adults; severe phenotypes	[4]
Step-up	Begin narrow, then escalate	Start with 1–2 foods (milk ± wheat) → add if no remission	Minimizes unnecessary restriction; child-friendly	May require multiple endoscopies; slower trigger identification	Pediatrics; nutritionally vulnerable	[9,24,26]

**Table 4 nutrients-17-03588-t004:** Remission rates and diagnostic performance of CRD-guided elimination versus empiric and elemental approaches.

Intervention Type	No. of Studies	Reported Remission Rate (%)	Diagnostic Accuracy (vs. Known Triggers)	Key References
CRD/test-directed diet	4	39.5–64	53–75	[5,25,28,30,32]
Empiric six-food elimination (SFED)	3	63.9–73	Not applicable	[1,4,29]
Elemental diet	4	90.8–96	Not applicable	[4,29,30]
Combination or immune-guided (T-cell/IgG4)	2	45–81.8	53–100	[25,31,32]

## Data Availability

No new data were created or analyzed in this study. Data sharing is not applicable to this article.

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
