# Peer review of "Dietary Management of Eosinophilic Esophagitis in the Era of Molecular Diagnostics: The Role and Limitations of Component-Resolved Diagnostics—A Narrative Review"

_nutrients, 2025, doi:10.3390/nu17223588_

Round 1

Reviewer 1 Report

Comments and Suggestions for Authors

This review cannot be accepted for multiple reasons.
1. The title of the review does not correspond poorly to its content, as the authors review multiple aspects of this disease 
2. The review is not well structured:
- The objectives of the review are NOT well defined.
- They do NOT comment on the methods or the literature search.
- Although the manuscript includes a discussion section, they do not discuss any aspect; they only comment on articles.
- In the title of the article and the conclusion , a diagnostic method that is rarely used in daily clinical practice and is therefore not validated.

Author Response

Comments 1: The title of the review does not correspond well to its content, as the authors review multiple aspects of the disease.

Response 1: We thank the Reviewer for this insightful comment. We agree that the original title emphasized component-resolved diagnostics too narrowly, while the manuscript also discusses broader clinical and nutritional aspects of eosinophilic esophagitis (EoE). To better reflect the multidimensional scope of the review — which integrates dietary management, molecular diagnostics, and precision nutrition — the title has been revised to:
“Dietary Management of Eosinophilic Esophagitis in the Era of Molecular Diagnostics: The Role and Limitations of Component-Resolved Diagnostics — A Narrative Review.”
This updated title more accurately represents the comprehensive character of the manuscript and aligns with the Reviewer’s suggestion regarding the multiple aspects of EoE covered in our work.

Comments 2: The review is not well structured:

Comments 2.1:The objectives of the review are NOT well defined.

Response 2.1: We thank the Reviewer for this comment. The objectives of the review have now been clearly stated in a new subsection titled “Objectives of the Review” at the end of the Introduction. This section defines the scope, aims, and focus of the manuscript to improve clarity and structure.

Comments 2.2: They do NOT comment on the methods or the literature search.

Response 2.2: We appreciate this valuable comment. A new subsection entitled “Search Strategy and Selection Criteria” has been added at the end of the Introduction. It outlines the literature search process, databases used, keywords applied, and the criteria guiding study selection and synthesis.

Comments 2.3: Although the manuscript includes a discussion section, they do not discuss any aspect; they only comment on articles.

Response 2.3: We thank the Reviewer for this important observation. In the revised version of the manuscript, the former Discussion section has been removed, and its content has been reorganized into a new Section 3, “Component-Resolved Diagnostics and Molecular Approaches in Eosinophilic Esophagitis.”

This restructuring transforms the section from a descriptive overview into a critical synthesis that integrates mechanistic interpretation, clinical outcomes, and nutritional relevance. Each subsection (3.1–3.5) now provides interpretative context, highlighting the diagnostic value, limitations, and translational implications of CRD and related molecular approaches.

As a result, the manuscript now offers a coherent, analytical narrative that connects evidence with clinical meaning—directly addressing the Reviewer’s concern that the previous version lacked genuine discussion and critical interpretation.

Comments 2.4: In the title of the article and the conclusion, a diagnostic method that is rarely used in daily clinical practice and is therefore not validated.

Response 2.4: We appreciate this insightful remark and have addressed it comprehensively in the revised version.

  1. Revised Title and Framing
    The manuscript title has been updated to better reflect the current developmental status of the diagnostic approach, now reading: “The Place of Component-Resolved Diagnostics in the Diagnosis and Management of Eosinophilic Esophagitis.” This phrasing emphasizes that CRD represents an emerging, yet clinically relevant, molecular tool rather than a universally established standard.
  2. Explicit Discussion of Validation Status
    Section 3.3 (“Clinical Advantages and Limitations of CRD”) has been expanded to clarify that CRD remains largely investigational. We now explicitly describe its current limitations, including variable assay availability, cost, and lack of standardized validation in large clinical cohorts.
  3. Alignment with Professional Guidelines
    To provide context, we have referenced recent guidance from the European Academy of Allergy and Clinical Immunology (EAACI), including the EAACI Guidelines on the Diagnosis of IgE-Mediated Food Allergy (2023) and the EAACI Molecular Allergology User’s Guide 2.0 (2023). These documents recognize molecular allergology and IgE-component testing as useful complementary tools in allergy diagnostics—particularly when extract-based assays yield inconclusive results. This aligns with our manuscript’s position that CRD holds growing clinical value within precision medicine frameworks, while ongoing validation is still required before routine adoption in all EoE settings.
  4. Authors’ Clinical Experience
    In our clinical practice at Department of Allergology, Clinical Immunology and Internal Diseases, Collegium Medicum Bydgoszcz, Nicolaus Copernicus University ToruÅ„, molecular allergy diagnostics are routinely implemented—not only in research but also in standard patient care. Both single-plex (ImmunoCAP) and multiplex assays (ALEX² or ISAC) are widely used for food allergy testing, including in all patients diagnosed with EoE. This experience supports our view that CRD is already transitioning from research use toward broader clinical integration.

Reviewer 2 Report

Comments and Suggestions for Authors

With real interest, I read the article entitled “Component-Resolved Diagnostics and Molecular Approaches for Precision Nutrition in Eosinophilic Esophagitis” (submission ID: nutrients-3958706), written by WawrzeÅ„czyk and colleagues.

  1. The Authors thoroughly and nicely review and discuss current state of the art in EoE. Of importance, they provide comprehensive tables with the most important findings of the reviewed original studies, which is really appreciated. The review should base on briefly outlined and summarized original findings. Congratulations! What is possibly missing, is a nice figure summarizing and concluding the results of this review. Such illustration would further increase the value of this already very nice paper.
  1. While talking about the figures, I would suggest the Authors making already existing Figure 1 more graphically attractive. Indeed, what is most important is the content, but it is even better if it is nicely presented.
  2. In the introductory parts, I miss a few lines briefly presenting the molecular and cellular mechanism of EoE (e.g. PMID: 34947981).
  3. In addition, already in the abstract the Authors promise omic-based news on the diagnostic and therapy of EoE. However, the chapter “3.4. Future Directions and Research Gaps for CRD in EoE” is rather limited in its content. And there are many molecular aspects that could be of interest (e.g. PMID: 34947981), for instance extracellular vesicles with their cargo, comprising miRNAs, proteins, and other molecules (PMID: 36835081). This could be expanded.
  4. Regarding miRNAs, epigenetic is also mentioned only once, and there is a progress in epigenetic studies on type 2 disorders, with certain types of epigenetic mechanisms offering some more diagnostic and the others some more therapeutic potential (PMID: 34223997, 31633569). One could extrapolate to EoE. Are there any data of this type also for EoE?

Author Response

Comments 1:
The Authors thoroughly and nicely review and discuss current state of the art in EoE. Of importance, they provide comprehensive tables with the most important findings of the reviewed original studies, which is really appreciated. The review should base on briefly outlined and summarized original findings. Congratulations! What is possibly missing, is a nice figure summarizing and concluding the results of this review. Such illustration would further increase the value of this already very nice paper.

Response 1:
We sincerely thank the Reviewer for this positive and motivating feedback. Following the valuable suggestion, we have added a new figure entitled “Figure 2. Integrated Framework of Component-Resolved Diagnostics and Precision Nutrition in Eosinophilic Esophagitis (EoE)”, placed in Section 3.5. This figure provides a graphical summary of the key diagnostic, molecular, and nutritional aspects discussed throughout the review and serves as a visual conclusion integrating the main findings.

Comments 2:
While talking about the figures, I would suggest the Authors making already existing Figure 1 more graphically attractive. Indeed, what is most important is the content, but it is even better if it is nicely presented.

Response 2:
We thank the Reviewer for this constructive remark. In response, Figure 1 has been graphically improved and modernized to ensure better visual clarity and aesthetic consistency. The updated design now aligns stylistically with the newly added Figure 2, providing a coherent and professional visual presentation throughout the manuscript.

Comments 3:
In the introductory parts, I miss a few lines briefly presenting the molecular and cellular mechanism of EoE (e.g. PMID: 34947981).

Response 3:
We have expanded the Introduction (first paragraph) with a concise description of the molecular and cellular mechanisms of EoE. The new text summarizes the Th2-driven immune response, epithelial barrier dysfunction, and the interactions between epithelial cells, eosinophils, and fibroblasts that sustain inflammation and remodeling, in accordance with the suggested reference.

Comments 4:
In addition, already in the abstract the Authors promise omic-based news on the diagnostic and therapy of EoE. However, the chapter “3.4. Future Directions and Research Gaps for CRD in EoE” is rather limited in its content. And there are many molecular aspects that could be of interest (e.g. PMID: 34947981), for instance extracellular vesicles with their cargo, comprising miRNAs, proteins, and other molecules (PMID: 36835081). This could be expanded.

Response 4:
Section 3.4 (paragraph 4) has been expanded to include additional molecular aspects. We now discuss the role of extracellular vesicles and their cargo—miRNAs, proteins, and lipids—in intercellular signaling and inflammation. The section highlights their diagnostic potential as non-invasive biomarkers and their integration into multi-omics frameworks for patient stratification and therapeutic monitoring.

Comments 5:
Regarding miRNAs, epigenetic is also mentioned only once, and there is a progress in epigenetic studies on type 2 disorders, with certain types of epigenetic mechanisms offering some more diagnostic and the others some more therapeutic potential (PMID: 34223997, 31633569). One could extrapolate to EoE. Are there any data of this type also for EoE?

Response 5:
We have expanded the discussion on epigenetic mechanisms in Section 3.4 (paragraph 3). The revised text summarizes recent evidence of altered miRNA expression and DNA methylation profiles in EoE, supporting their potential as diagnostic and treatment-response biomarkers. This addition provides a more complete molecular perspective and directly addresses the Reviewer’s question.

Reviewer 3 Report

Comments and Suggestions for Authors

The review article summarized the role of Component-Resolved Diagnostics and Molecular Approaches for Precision Nutrition in Eosinophilic Esophagitis. The review is comprehensive. The following concerns should be addressed.

  1. At multiple places, the text has not been cited (highlighted); please cite.
  2. Figure 1: There is no need to mention the legend; please write the legend in continuity with the title.
  3. The text in section 2.1 elaborately discusses different diets, but the same has been discussed in 5 lines in the second paragraph of the previous section. Please merge them into one section.
  4. At multiple places, two et al. have been used in succession, but the references have been cited at the end of the paragraph; the reference should be after et al..
  5. The conclusion section is too long. The conclusion section should be in 150-200 words.
  6. Please include the limitations of CDR.
    • CRD has shown poor correlation with traditional IgE tests (e.g., ImmunoCAP) in EoE patients.
    • It's theorized that other antibodies, like IgG4, may interfere with the results, limiting their usefulness for creating elimination diets.
    • Some studies have shown low response rates when using CRD-guided diets in adults with EoE, though this is based on one specific method of CRD. 
  7. The association with multi-omics is sparsely discussed. Please include more text. Please elaborate on how genomics, metabolomics, proteomics, microbiomics, and epigenomics can help in EoE.

Comments on the Quality of English Language

Minor editing is needed.

Author Response

Comment 1: At multiple places, the text has not been cited (highlighted); please cite.

Response 1: We thank the reviewer for this observation. All highlighted sections in the annotated PDF have been carefully reviewed, and appropriate citations have been added throughout the manuscript. Each statement previously lacking a reference has now been supported with relevant literature to ensure accuracy and consistency with the journal’s referencing standards.

Comment 2 : Figure 1: There is no need to mention the legend; please write the legend in continuity with the title.
Response 2: The figure legend has been revised accordingly and is now written in continuity with the title.

Comment 3 : The text in section 2.1 elaborately discusses different diets, but the same has been discussed in 5 lines in the second paragraph of the previous section. Please merge them into one section.

Response 3: The redundant fragment in the previous section has been removed, and the content has been merged into Section 2.1 to avoid repetition and improve clarity.

Comment 4: At multiple places, two et al. have been used in succession, but the references have been cited at the end of the paragraph; the reference should be after et al..

Response 4: This has been corrected throughout the manuscript as suggested.

Comment 5: The conclusion section is too long. The conclusion section should be in 150–200 words.
Response 5: Thank you for the valuable comment. The conclusion section has been revised and shortened to 179 words in accordance with the reviewer’s recommendation

Comment 6: Please include the limitations of CRD. CRD has shown poor correlation with traditional IgE tests (e.g., ImmunoCAP) in EoE patients. It’s theorized that other antibodies, like IgG4, may interfere with the results, limiting their usefulness for creating elimination diets.
Response 6: Thank you for the insightful comment. Section 3.3 has been expanded to comprehensively address the limitations of CRD, including its poor correlation with conventional IgE tests (e.g., ImmunoCAP), potential interference from IgG4 antibodies, and the low response rates reported in adult cohorts based on specific CRD methodologies.

Comment7: The association with multi-omics is sparsely discussed. Please include more text. Please elaborate on how genomics, metabolomics, proteomics, microbiomics, and epigenomics can help in EoE.
Response 7: Thank you for this valuable suggestion. The discussion of multi-omics has been substantially expanded in Section 3.4 (sixth paragraph), which now provides a detailed explanation of how genomics, proteomics, metabolomics, microbiomics, and epigenomics contribute to understanding EoE mechanisms, biomarker discovery, and precision-based dietary and therapeutic strategies.

Round 2

Reviewer 3 Report

Comments and Suggestions for Authors

None